# Thermal Characteristics Simulation of an Energy-Conserving Facade: Water Flow Window

**Yuanli Lyu [1,2,\*], Sihui Chen [1], Can Liu [1], Jun Li [1], Chunying Li [3] and Hua Su [1]**

[1] Department of Civil, Architecture and Environment, Xihua University, Chengdu 610097, China; 212020085900094@stu.xhu.edu.cn (S.C.); xrak@cdxrec.com (C.L.); 21201908513027@stu.xhu.edu.cn (J.L.); xhsuhua@mail.xhu.edu.cn (H.S.)

[2] Key Laboratory of Fluid and Power Machinery, Ministry of Education, Xihua University, Chengdu 610097, China

[3] BenYuan Design and Research Center, School of Architecture and Urban Planning, Shenzhen University, Shenzhen 518060, China; lichunying@szu.edu.cn

\* Correspondence: 1220170064@mail.xhu.edu.cn

**Abstract:** In this paper, a 3D numerical simulation was completed to explore the thermal characteristics in a water flow window in-depth. CFD was used to carry out the analysis on top of successful validation. By changing the solar intensity, water supply temperature and velocity, the temperature distribution and flow field in the window cavity, as well as the water heat gain, were analyzed and compared. This is meaningful for improving the energy-conserving performance in building applications. Simulation results reveal that the variation of solar intensity and water supply temperature affects directly the temperature distribution and the water heat gain but has little impact on the overall velocity field. Local vortices are generated in the window cavity, and their formation and location are largely affected by the varied temperature rise in the water layer. The water heat gain increases and then decreases with the increase in water supply velocity. In addition, a large-enough water supply velocity can disorder the uniform upward flow. These are detrimental to effective thermal extraction. Therefore, in practical application, the vortex should be eliminated, and the flow velocity should be determined properly to maximize the water heat gain.

**Keywords:** water flow window; temperature distribution; flow field; energy conservation

## 1. Introduction

According to the International Energy Agency (IEA) survey in 2018, buildings are responsible for approximately 36% of the final energy used, and higher percentages are observed in developed countries, for example, 38% in Europe and 39% in the U.S. [1]. In 2018, China's building energy consumption accounted for 37% of the total social energy consumption [2]. Among the building facade elements, windows account for the largest energy loss due to their notably high heat transfer coefficients [3]. To reduce building energy consumption, passive adaptation design is one of the effective approaches to cut the heat transmission through the building envelope [4,5]. This paper pays much attention to the weak insulation component, the window [6]. The water flow window [7] is a multi-glazing system with one or more flowing water layers in the window cavities. In addition to its excellent thermal characteristics [8], the water flow window can also absorb and utilize part of the incident solar radiation. Therefore, the water flow window works as a building-integrated cooling/heating radiator or solar collector [9,10]. In addition, the quality of sunlight transmission is not affected because of the good transparency of the water layer [11].

Its energy-saving potential has been extensively verified. For a non-air-conditioned room in Spain, the use of a water flow window lowered the room temperature by 18 K [12]. Taking the Hong Kong Fitness Club as an example, Chow's research found that, compared with traditional single- and double-layer windows, the annual thermal transmissions

through the windows were reduced by 52% and 32%, respectively [9]. In cold winter applications, antifreeze was added to the water for safe operation, and the annual thermal efficiency of the natural circulation water flow window varied between 8.58% and 15.62% with climate change [13]. Supplying warm water to the window was another alternative; with this active approach, room temperature was elevated, and zero heat loss could be achieved under proper operation [14,15]. Compared with natural circulation (Figure 1b), thermal performance of the window with forced circulation of cold feed water (Figure 1a) could be improved.

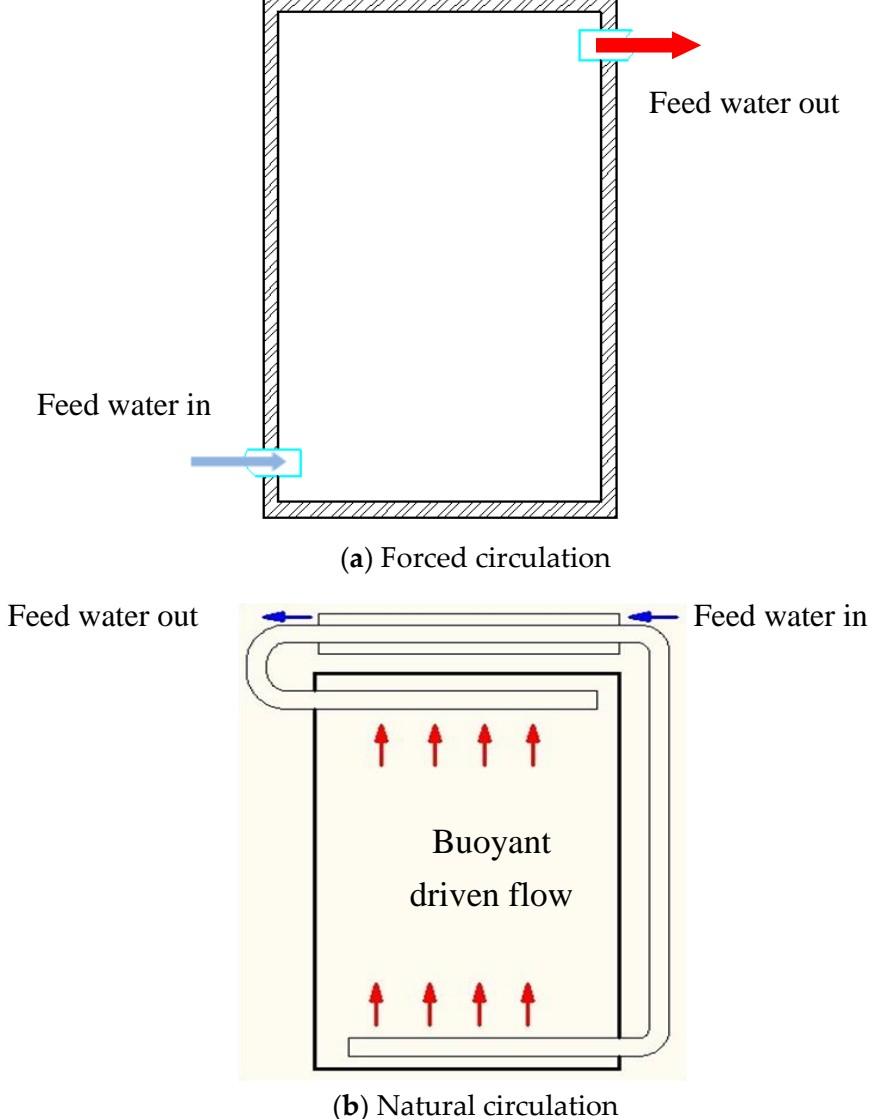

**Figure 1.** Schematic structure of water flow window [14].

Thermal performance of the window system [16] was found affected by weather conditions [8] and water flow [17], as well as the proper determination of window schematic structure [18,19] and glazing material [9,20]. Higher thermal efficiency was achievable under large solar intensity and low ambient temperature conditions. Varying the water flow velocity helped to regulate the thermal performance, and its impact was found more significant when the velocity was less than 0.01 m/s [17]. This was completed by comparing the thermal transmission through the window and system thermal efficiency. From the above, considering the influence of external and internal factors, energy-saving performance analysis of the water flow window has been widely completed. The previous studies focused more on the impact of the water flow window application on the surrounding thermal

environment. However, they failed to display the detailed temperature and velocity distribution characteristics, which are determinants of the thermal performance of liquid flow in the window cavity. In this paper, the thermal characteristics of the water flow window are studied through CFD simulation under various conditions. In-depth understanding of the temperature and velocity distribution as well as the heat exchange mechanism in the flow field is meaningful for optimizing its energy-conserving performance.

CFD that uses advanced computer technology as a means, with the help of advanced discrete mathematics to solve nonlinear simultaneous differential equations of mass, energy, momentum and custom scalar, has been successfully used in the temperature and flow pattern analysis of the complex multi-layer envelope [21]. It was used to analyze the radiant environment to study the impact of windows with near-infrared retro-reflective film on the outdoor thermal environment [22]. Given a set of outdoor wind conditions, a numerical analysis of the integrated thermal performance of ventilation windows was carried out with CFD [23]. Two-dimensional [24] and three-dimensional [25] simulations were also completed to explore the temperature and air flow distribution in the cavity of double-glazed windows and curtain walls. The results reveal the detailed distribution of temperature and velocity in the flow field accurately and intuitively. Therefore, this article investigates the temperature distribution and velocity field of the water layer in the window cavity with CFD. It has been proven to be accurate enough to predict the steady-state performance of the water flow window [26].

In the current study, the effects of solar radiation intensity, supply water temperature and velocity on thermal characteristics were analyzed. The ANSYS Workbench software with the integrated functions of model creation, mesh generation and fluent simulation was used to conduct the study.

## 2. Numerical Method and Validation

### 2.1. Mathematical Models

The mass, momentum and energy balance equations for CFD analysis are given below:

(1)　Mass balance equation

$$\nabla \cdot \vec{u} = 0 \tag{1}$$

where $\vec{u}$ represents the flow velocity vector, m/s.

(2)　Momentum balance equation

$$\frac{\partial}{\partial t}\left(\rho \vec{u}\right) + \nabla \cdot \left(\rho \vec{u}\vec{u}\right) = -\nabla p + \nabla \cdot \left(\overline{\overline{T}}\right) + \rho \vec{g} \tag{2}$$

where $\rho \vec{g}$ represents the gravitational force driving the buoyancy flow, and $\overline{\overline{T}}$ is the stress tensor given by

$$\overline{\overline{T}} = \mu \left[\left(\nabla \vec{u} + \nabla \vec{u}^T\right)\right] \tag{3}$$

where $\mu$ is the molecular viscosity, kg/(m·s).

(3)　Energy balance equation

$$\frac{\partial}{\partial t}(\rho C T) + \nabla \cdot \left(\vec{u}(\rho C T + p)\right) = \nabla \cdot \left(k \nabla T + \left(\overline{\overline{T}} \cdot \vec{u}\right)\right) + S \tag{4}$$

in which the first two terms at the right side represent energy transfer owing to thermal conduction and viscous dissipation, respectively, and $S$ indicates the volumetric heat source, W/m$^3$, which is the direct solar absorption for the water layer. In addition, $C$ represents the specific heat capacity, J/(kg·K); $p$ and $T$ are the working pressure (Pa) and temperature (K); and $k$ is the thermal conductivity, W/(m·K).

In CFD simulation, the convective heat transfer coefficients at the inner and outer glazing surfaces are determined by the formula given below [27]:

$$H_b = 2.8 + 3V_{wind} \qquad (5)$$

where $H_b$ represents the heat transfer coefficient, $W/(m^2 \cdot K)$, and $V_{wind}$ is the air velocity, m/s.

Relative error (RE) and relative mean error (RME) calculated by the following equations are used for accuracy analysis in the validation study.

$$RE = \frac{|T_{sim} - T_{exp}|}{T_{exp}} \qquad (6)$$

$$RME = \frac{\sum_1^N RE}{N} \qquad (7)$$

In that, $T_{sim}$ and $T_{exp}$ represent the simulation and experimental testing results, respectively, and $N$ is the testing number during the testing duration.

The water heat gain is calculated for thermal performance analysis with the formula given below:

$$Q_w = C_w \dot{m}_w (T_{outlet} - T_{inlet}) \qquad (8)$$

where $Q_w$ refers to the water heat gain, W; $C_w$ is the specific heat capacity of water, $J/(kg \cdot K)$; $\dot{m}_w$ is the mass flow rate of the water stream, kg/s; $T_{inlet}$ and $T_{outlet}$ are the temperatures of the water stream at window inlet and outlet, respectively, °C.

### 2.2. Validation Study

### 2.2.1. Experimental Setup

In this paper, the accuracy of the numerical method was verified by comparing it to the experimental testing results. The experiment was completed in Chengdu, Sichuan Province, from 9:00 to 17:00 on 21 August 2020. The window prototype under investigation was 0.96 m (W) × 0.59 m (H) × 0.032 m (T) and was composed of two pieces of clear glazing at 10 mm and a 12 mm thick flowing water layer. The experimental setup and window structure is shown in Figure 2. The window was installed on the wall of the experimental cabin due south to receive more solar radiation. A water tank was arranged outside the experimental cabin, and the water from the municipal pipe network was transported to the window inlet by a water pump. The water tank and the pipeline were insulated to reduce water pre-heating in the pipeline. Low-temperature municipal water was supplied to the bottom of the window cavity and extracted from the water outlet at the top. A flowmeter was installed at the inlet to control and measure the water flow rate. Since the experiment focused mainly on the thermal performances of the water flow window, the water stream flowing out of the window cavity was drained during the testing. In practical application, the water flow window can be used as a water pre-heating device in health clubs, sports centers and other places with stable indoor temperature and hot water demand. The thermophysical and optical properties of the above clear glazing at normal incidence are given in Table 1.

**Table 1.** Thermophysical and optical properties of 10 mm clear glazing at normal incidence.

| Optical Property | | Thermophysical Property | |
|---|---|---|---|
| Transmittance | 0.672 | Density $(kg/m^3)$ | 2515 |
| Reflectance | 0.069 | Conductivity $W/(m \cdot K)$ | 1 |
| Absorptance | 0.259 | Thermal capacity | |
| Emissivity | 0.84/0.84 | $J/(kg \cdot K)$ | 810 |

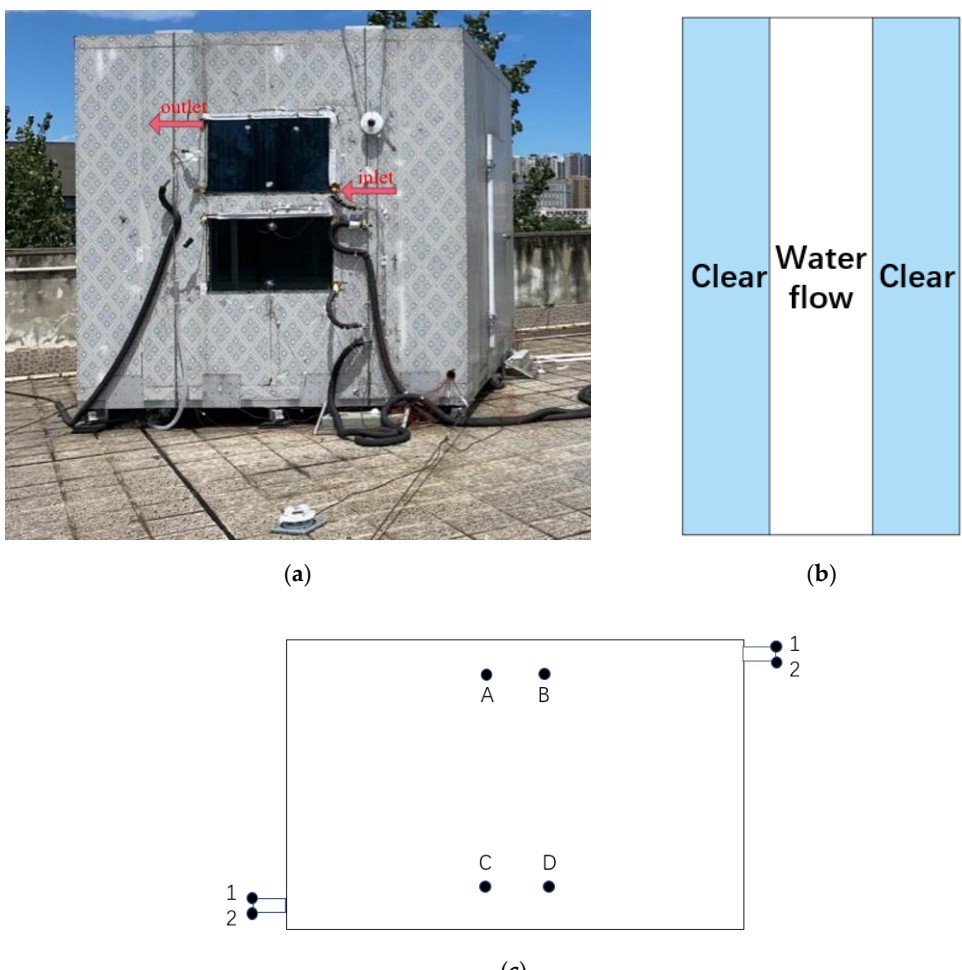

**Figure 2.** Experimental setup of water flow window. (**a**) Outlook of the test chamber; (**b**) side view of water flow window; (**c**) distribution of temperature measuring points.

In the experimental testing, two pyranometers were used to measure the total solar radiation; temperature and humidity recorders were used for ambient temperature measurement inside and outside the experimental cabin, and T-type thermocouples were used for glazing surface temperature and water inlet and outlet temperature measurements. In particular, two measuring points were arranged at the inlet and outlet, and they were completely covered by the water flow to reduce measurement error. The distribution of measuring points on the glass surface is shown in Figure 2c; the measurement points were arranged 5 cm from the top and bottom edges. In order to avoid the influence of direct solar radiation on the measurement results, the thermocouples at the outer glazing surface were wrapped with tin foil. The air conditioner was turned on one hour in advance to maintain the indoor temperature at 298.15 ± 1 K. The incident solar radiation, room and ambient temperatures, glass surface temperatures, flow rate and temperatures of the water stream at window inlet and outlet were measured. They were recorded at 10 s intervals; the hourly averaged results are given in Table 2. Among them, the solar intensity, indoor and outdoor ambient temperatures, flow rate and temperature of supply water were used as simulation inputs. The accuracy of the numerical method was verified by comparing the calculated glazing surface temperatures and water outlet temperature with the testing results as given in Table 3.

**Table 2.** Testing results used as simulation input.

| Time (h) | Solar Intensity (W/m$^2$) | $T_{indoor}$ (K) | $T_{outdoor}$ (K) | $T_{inlet}$ (K) | Inlet Velocity (m/s) |
|---|---|---|---|---|---|
| 9:00 | 506.83 | 297.50 | 303.23 | 298.34 | 0.13 |
| 10:00 | 693.94 | 297.59 | 306.33 | 298.88 | 0.13 |
| 11:00 | 815.65 | 297.78 | 308.04 | 299.32 | 0.13 |
| 12:00 | 883.93 | 297.94 | 312.91 | 299.67 | 0.13 |
| 13:00 | 842.08 | 298.08 | 313.21 | 299.76 | 0.13 |
| 14:00 | 733.79 | 298.04 | 313.04 | 299.75 | 0.13 |
| 15:00 | 591.67 | 297.96 | 312.41 | 299.39 | 0.13 |
| 16:00 | 289.03 | 297.96 | 309.56 | 298.89 | 0.13 |
| 17:00 | 170.09 | 297.73 | 306.82 | 298.36 | 0.13 |

**Table 3.** Glazing surface temperatures and water temperature at window outlet.

| Time (h) | $T_{inner-glazing}$ (K) | $T_{outer-glazing}$ (K) | $T_{outlet}$ (K) |
|---|---|---|---|
| 9:00 | 299.56 | 299.94 | 299.71 |
| 10:00 | 300.96 | 301.99 | 301.26 |
| 11:00 | 301.96 | 303.70 | 302.64 |
| 12:00 | 302.56 | 304.55 | 303.49 |
| 13:00 | 302.84 | 304.84 | 303.99 |
| 14:00 | 302.57 | 304.53 | 303.78 |
| 15:00 | 301.80 | 303.63 | 302.84 |
| 16:00 | 300.83 | 301.97 | 301.83 |
| 17:00 | 299.69 | 300.45 | 300.36 |

### 2.2.2. Validation Simulation

In the validation study, the physical model of the window prototype was created in the first place, and ICEM was used to mesh the computational area with a structured hexahedron. To deal with the complicated heat and mass exchange at window inlet and outlet, the number of grids at the corresponding structures was increased. With grid numbers of 305,000, the flow field and temperature distribution were found essentially correct. Further increase in the grid numbers contributed little to the change of simulation results. Therefore, the accuracy of the simulation results was considered acceptable at grid numbers of 305,000.

The modeling of heat transfer in the water flow window was coupled with the calculation of heat conduction, convection and radiation. The influence of solar radiation through the outer glazing on the heat gain of the water layer and the inner glazing pane is also significant. The discrete ordinates (DO) method, which is able to predict the radiation heat transfer in the semi-transparent medium, was chosen to solve the radiation transfer equation (RTE). The discretized angles were grouped, and the equations solved by the finite volume method were obtained. The laminar flow model was selected in the current study with a supply water velocity of 0.13 m/s.

Following this, the boundary conditions were defined. Window frames were considered to be adiabatic by assuming that perfect insulation was applied. The water inlet was defined as velocity-inlet at 0.13 m/s, and the outlet was fully-developed outflow. Non-slip boundary conditions were applied to all the surfaces in the calculation domain. The interface between the glazing surface and the water layer was defined as the mixing boundary. The convective heat transfer coefficients at the inner and outer glazing surfaces were 4.3 W/(m$^2$·K) and 6.7 W/(m$^2$·K) determined by the above Equation (5).

In the simulation, SIMPLEC algorithm was used to couple the velocity and pressure. In order to predict the effect of buoyancy, the body-force-weighted method was used to discretize the pressure. The convection and diffusion terms were discretized by the second-order upwind difference scheme. Convergence could be achieved when the user-defined

error tolerance was reached. In this study, it was $10^{-6}$ and $10^{-7}$, respectively, for velocity and energy.

For validation, the window surface temperatures and water temperature at window outlet from numerical simulation and experimental testing were compared. The corresponding REs and RMEs calculated with Equations (6) and (7) were used for accuracy analysis, and the results from 9:00 to 17:00 at an hourly averaged scale are given in Table 4.

**Table 4.** Relative error and relative mean error of glazing surface temperatures and water outlet temperature.

| Time (h) | $RE_{inner-glazing}$ (%) | $RE_{outer-glazing}$ (%) | $RE_{outlet}$ (%) |
|---|---|---|---|
| 9:00 | 3.75 | 4.39 | 3.36 |
| 10:00 | 4.89 | 7.19 | 3.66 |
| 11:00 | 4.39 | 8.66 | 3.10 |
| 12:00 | 2.53 | 7.24 | 0.36 |
| 13:00 | 3.27 | 7.93 | 1.74 |
| 14:00 | 3.31 | 7.82 | 2.50 |
| 15:00 | 3.77 | 7.99 | 3.17 |
| 16:00 | 4.78 | 7.18 | 5.71 |
| 17:00 | 3.74 | 5.16 | 4.43 |
| RME (%) | 3.83 | 7.06 | 3.11 |

The REs of inner and outer glazing surface temperatures are in the ranges of 2.53–4.89% and 4.39–8.66%, and the corresponding RMEs are 3.83% and 7.06%, respectively. The RE of water outlet temperature varies from 0.36% to 5.71% with an RME of 3.11%. REs of the outer glazing surface temperature are slightly larger; this can be caused by the calculation with a constant convective heat transfer coefficient at the outer glazing surface. The calculated values from the numerical simulation are essentially consistent with the experimental testing results. The acceptable [28] REs obtained in the present study indicate that the CFD simulation method in use is accurate enough for the thermal performance prediction of the water flow window.

## 3. Numerical Simulation under Varied Operation Conditions

On top of the successful validation, CFD was further used to evaluate the thermal characteristics of the water flow window under various operation conditions. The dimensions of the window under investigation were 1.2 m high, 0.8 m wide, and the water layer in between was 20 mm thick at a solar absorption coefficient of 0.187 [18]. The glazing panes used were 8 mm absorptive glazing. The water inlet and outlet were of the same dimensions of 12 mm × 40 mm. The 3D physical model of the window under investigation is given in Figure 3.

Similar to the validation study, the simulation domain was meshed with ICEM, and the number of grids at the window inlet and outlet was increased to improve the accuracy of calculation. The details of the local grid structure at window inlet are shown in Figure 4. Grid independence was verified with grid numbers of 150,000, 240,000, 430,000, 670,000 and 900,000. When the number of grids was 670,000, the physical phenomenon presented by the simulation results was essentially correct, and the simulation results changed little with a further increase in the grid numbers. Therefore, the model with grid numbers of 670,000 was used for simulation.

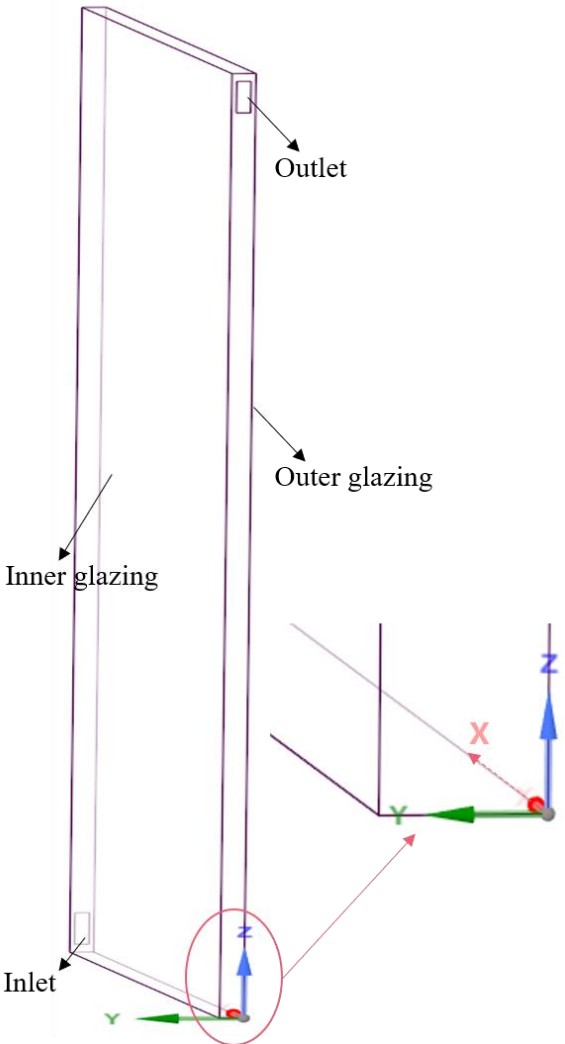

**Figure 3.** The 3D physical model of water flow window.

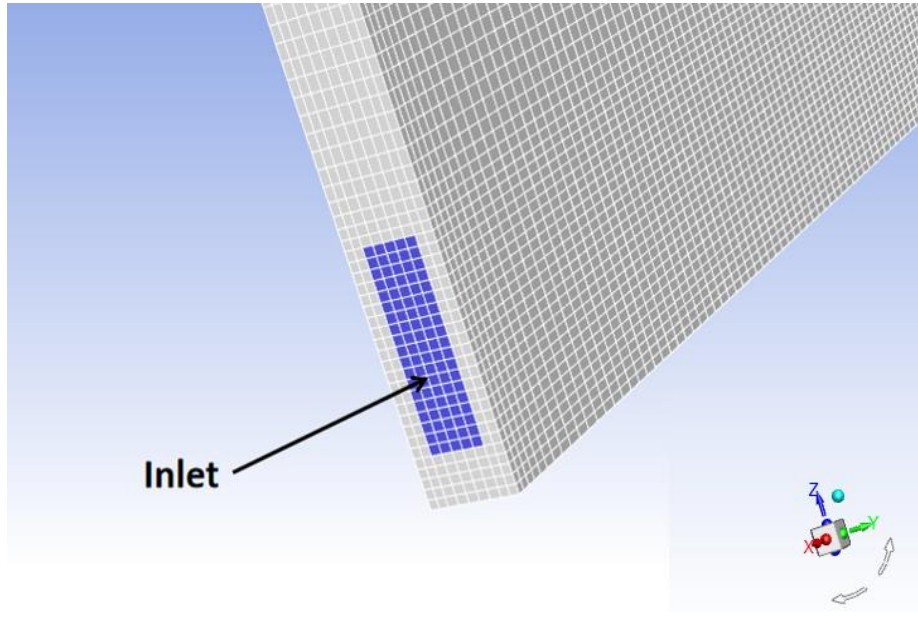

**Figure 4.** Local mesh structure at window inlet.

The discrete ordinates (DO) model was selected for solar radiation modeling, and laminar or turbulent flow models were used depending on the different supply water velocities. With supply water at high velocity, the flow was mainly affected by forced flow at the window inlet. Together with the limitation of the solid surface in the window cavity, insufficient flow caused local turbulence in the flow field; the standard k-ε model was then adopted, as indicated in Table 5. In the K-epsilon model, the near-wall treatment uses Standard Wall Function, and the turbulence intensity is 5%. The room and ambient temperatures were set at 298 K and 304 K, respectively. The convection heat transfer coefficients at the inner and outer glazing surfaces were determined by Equation (5) at values of 4.3 W/(m²·K) and 6.7 W/(m²·K). The convergence criteria were the same as in the validation study.

**Table 5.** Cases under investigation.

| Case | $T_{inlet}$(K) | $V_{Inlet}$(m/s) | Solar Intensity (W/m²) | Viscous Model |
|------|------|------|------|------|
| 1 | 291 | 0.033 | 400 | Laminar |
| 2 | 291 | 0.033 | 600 | Laminar |
| 3 | 291 | 0.033 | 800 | Laminar |
| 4 | 285 | 0.033 | 600 | Laminar |
| 5 | 298 | 0.033 | 600 | Laminar |
| 6 | 291 | 0.017 | 600 | Laminar |
| 7 | 291 | 0.083 | 600 | K-epsilon |
| 8 | 291 | 0.17 | 600 | K-epsilon |

The impacts of variation of solar intensity, supply water temperature and velocity on the thermal characteristics were investigated. Solar intensities at high, mild and low levels (800 W/m², 600 W/m² and 400 W/m²) were taken into consideration. The determination of water supply temperature considered three values. The maximum one was 298 K (the same as room temperature), and the others were the same as the temperature of urban supply water in the cooling season, with the minimum of 285 K and the maximum of 291 K. The determined water supply velocities were 0.017 m/s, 0.033 m/s, 0.083 m/s and 0.17 m/s, corresponding to flow rates of 0.008 kg/s, 0.016 kg/s, 0.04 kg/s and 0.08 kg/s. The cases under investigation are summarized in Table 5. The one with a mild solar intensity of 600 W/m², supply water temperature of 291 K and velocity of 0.033 m/s was taken as the base case (Case 2).

## 4. Results and Discussion

### 4.1. The Impact of Solar Intensity

Figure 5 shows the temperature distribution in the window cavity at Y = 0.01 m, with a constant velocity of 0.033 m/s and temperature of 291 K for supply water. The effects of solar intensity at 400 W/m², 600 W/m² and 800 W/m² on the temperature distribution of the water layer are compared.

The supplied water absorbs solar thermal energy in the cavity, and its temperature rises gradually. Under the influence of buoyancy, a temperature rise along the height direction is demonstrated in Figure 5. With the increase in solar intensity, the heat absorption of the fluid in the cavity is enhanced, and a larger temperature rise is then observed. In Case 1, the solar radiation is at the minimum level, and thermal energy extracted by the water stream in the cavity is then the least. As a consequence, the temperature rise is of the smallest value. Its maximum temperature at the top of the window cavity is 294.7 K. The solar intensity of Case 3 is 800 W/m², which is the maximum among the three cases. The water temperature at the top of the window cavity is then the highest among the three cases, reaching 296.2 K, which is 1.5 K higher than that of Case 1.

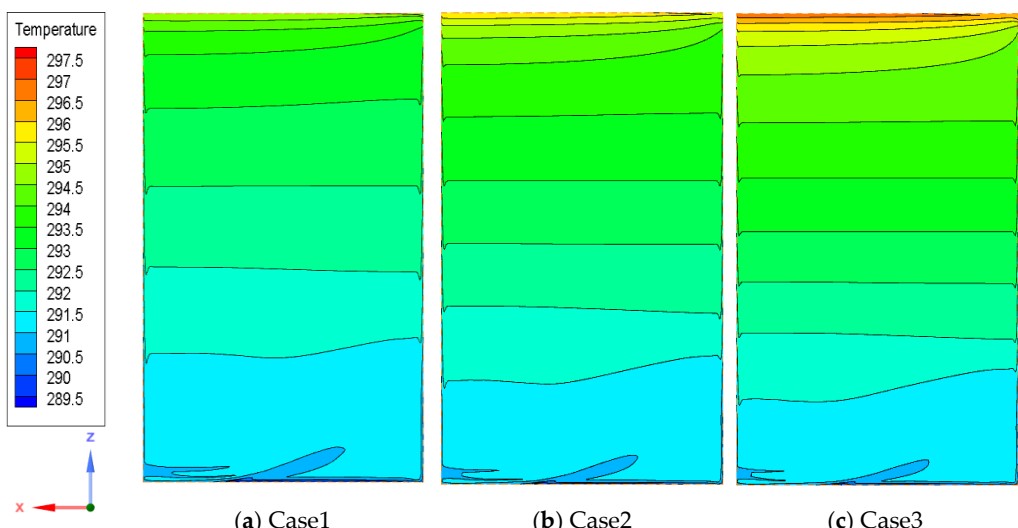

**Figure 5.** Temperature distribution at Y = 0.01 m.

Table 6 lists the water temperatures at window inlet and outlet, the water heat gain and the surface temperatures of the inner and outer glazing panes. With the increase in solar intensity by 200 W/m$^2$, the water temperature at the outlet increases by about 0.75 K, and the corresponding water heat gain increases by about 49 W. Due to the influence of direct solar radiation, the surface temperature of the outer glazing pane is higher than that of the interior glazing pane. For every increase of 200 W/m$^2$ of solar radiation, the temperature rise of inner glazing is as small as 0.36 K, and that of outer glazing is around 0.5 K. The surface temperature of the interior glazing pane can be kept at a relatively low level with the varied solar intensity, in the range of 292.41 K to 293.12 K for the three cases. This helps to maintain the stability of the indoor thermal environment even in strong solar radiation conditions. At the same time, the discomfort caused by the asymmetrical radiation near the window area can be reduced.

**Table 6.** Temperature of water stream and glazing surfaces and water heat gain.

| Case | $T_{Inlet}$ (K) | $T_{outlet}$ (K) | $Q_w$ (W) | $T_{inner-glazing}$ (K) | $T_{outer-glazing}$ (K) |
|------|-----------------|------------------|-----------|-------------------------|-------------------------|
| 1 | 291 | 293.71 | 180.92 | 292.41 | 293.10 |
| 2 | 291 | 294.47 | 229.84 | 292.76 | 293.60 |
| 3 | 291 | 295.21 | 278.79 | 293.12 | 294.09 |

The overall velocity distribution of Cases 1–3 at Y = 0.01 m is given in Figure 6, and a non-significant difference is observed. Figure 7 shows the streamline of the three cases at X = 0.4 m. Local vortices are generated near the window inlet and outlet for all three cases, as can be seen in Figure 7a. They are caused by the incoming flow and outflow with large velocity and the limitation of gravity and cavity space on fluid flow. However, additional counter-clockwise vortices are observed in the middle of the window cavity in Case 2 and Case 3, with the locations indicated in Figure 7b,c. The backflow vortex in Case 3 is located at a lower position under the influence of temperature difference between the water layer near the glazing surface and in the middle.

Figure 8 shows the temperature distribution at X = 0.4 m for more intuitive illustration. The outer glazing exposed directly to the sunlight tends to have a higher overall temperature. The temperature of the inner glazing pane is affected by direct solar absorption and convective heat exchange with room space, and both of them are higher than the temperature of supply water. As a consequence, water near the glass surface is heated to a higher temperature compared to that in the middle. Driven by this temperature difference, the counter-clockwise vortices are formed in the water cavity, as indicated in Figure 7b,c.

With a solar intensity of 800 W/m$^2$, the large-enough temperature difference, which drives the water circulation, is observed at a lower position as indicated in Figure 8c. Thus, the location of the vortex is lower for Case 3 (Figure 7c) compared to Case 2 (Figure 7b).

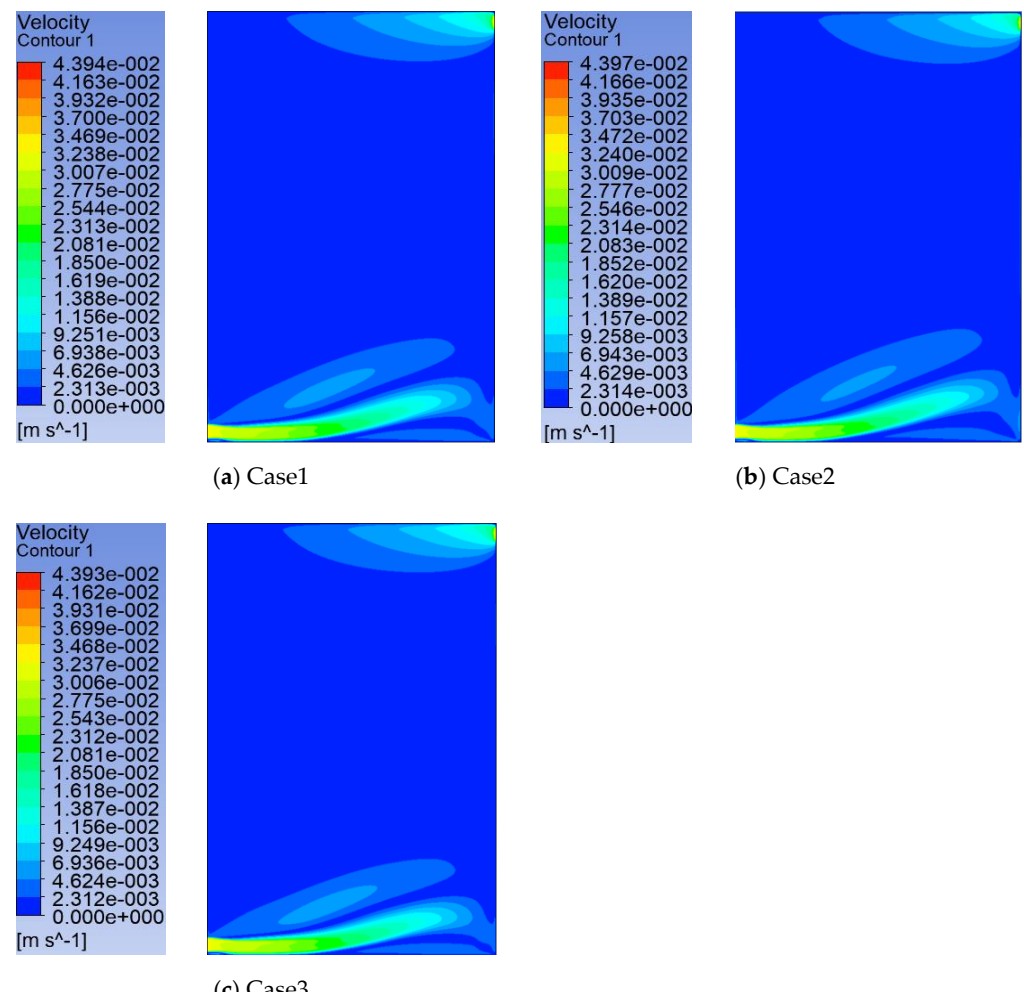

(**a**) Case1    (**b**) Case2

(**c**) Case3

**Figure 6.** Velocity distribution at Y = 0.01 m.

The average water temperatures at Y = 0.003 m (near inner glazing), Y = 0.01 m (in the middle) and Y = 0.017 m (near outer glazing) are listed in Table 7. With the increase in solar radiation, the average water temperature increases, and this is true for all three surfaces. There is a maximum average temperature at Y = 0.017 m under the direct heating of solar radiation. The temperature rise of water in the middle cavity is of smaller magnitude, and the corresponding average temperature is the minimum for all three cases. This is consistent with the results in Figures 7 and 8.

**Table 7.** Average temperatures of different section surfaces.

| Case | $T_{y=0.003m}$ (K) | $T_{y=0.01m}$ (K) | $T_{y=0.017m}$ (K) |
|---|---|---|---|
| 1 | 292.28 | 292.18 | 292.44 |
| 2 | 292.62 | 292.53 | 292.78 |
| 3 | 292.96 | 292.88 | 293.12 |

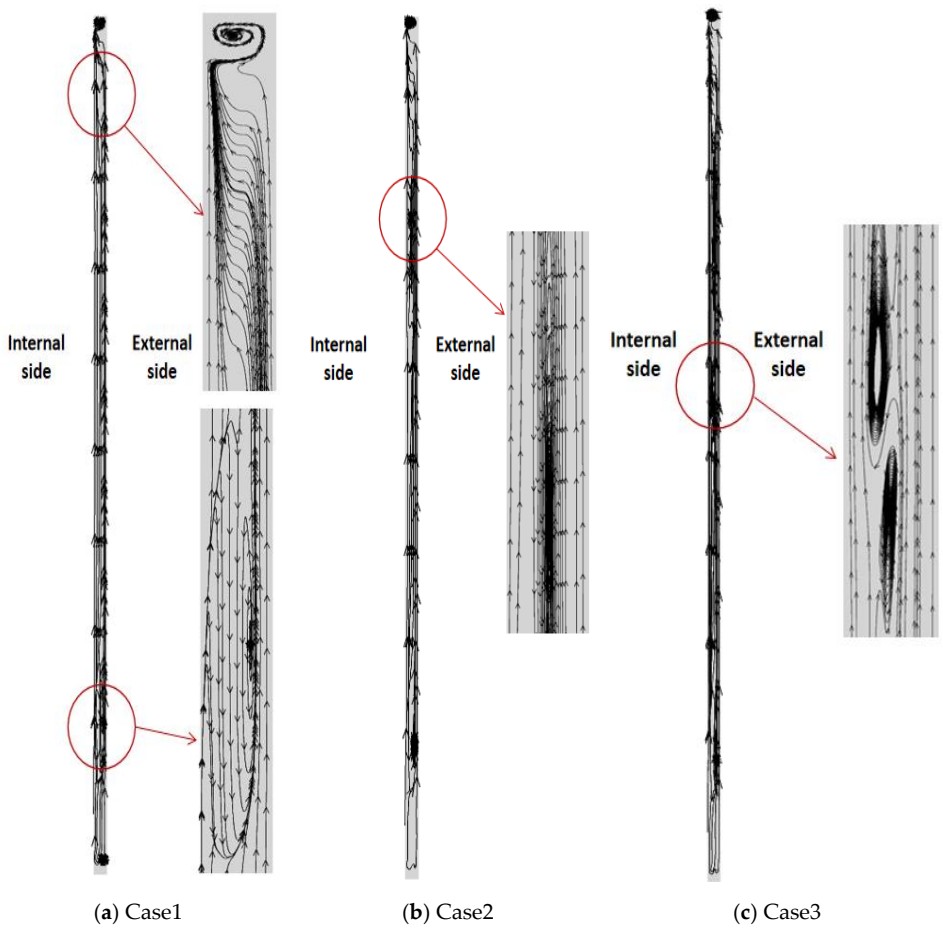

(**a**) Case1　　　　　　　　(**b**) Case2　　　　　　　　(**c**) Case3

**Figure 7.** Streamline of water flow at X = 0.4 m.

The average water flow velocities at the above-mentioned section surfaces are compared in Table 8. Affected by the higher temperature as given in Table 7, the average velocity is the maximum at the section surface of Y = 0.017 m, since the upward buoyancy-driven flow is accelerated. Similarly, the fluid flow at Y = 0.003 m is affected significantly by the convective heat exchange with the indoor environment. As a consequence, the velocity at the Y = 0.003 m section is also high. The water temperature in the middle cavity is the lowest, since a backflow vortex (Figure 7) is formed because of the temperature difference between the water layer near the glazing surfaces and in the middle as shown in Figure 8.

From the above, when the solar intensity increases by 200 W/m², the water heat gain increases by about 49 W; this can effectively reduce the room heat gain through the window. At the same time, the heated circulating water can be introduced into the building's hot water system. Therefore, in areas with strong solar radiation, the water flow window can be used as an effective water pre-heating device, which can not only realize building energy saving but also realize solar energy utilization, consequently reducing carbon emissions during both energy generation and building operation processes. However, it is also worth noting that vortices generated in the water cavity result in heat storage and reduce heat removal. This must be considered to improve the thermal performance of the water flow window in practical application.

**Table 8.** Z axis average velocity in the cavity.

| Case | $V_{y=0.003m}$(m/s) | $V_{y=0.01m}$(m/s) | $V_{y=0.017m}$(m/s) |
|------|------|------|------|
| 1 | 0.0005282 | 0.0001591 | 0.002674 |
| 2 | 0.0005202 | 0.0000439 | 0.002859 |
| 3 | 0.0005102 | −0.0000251 | 0.002977 |

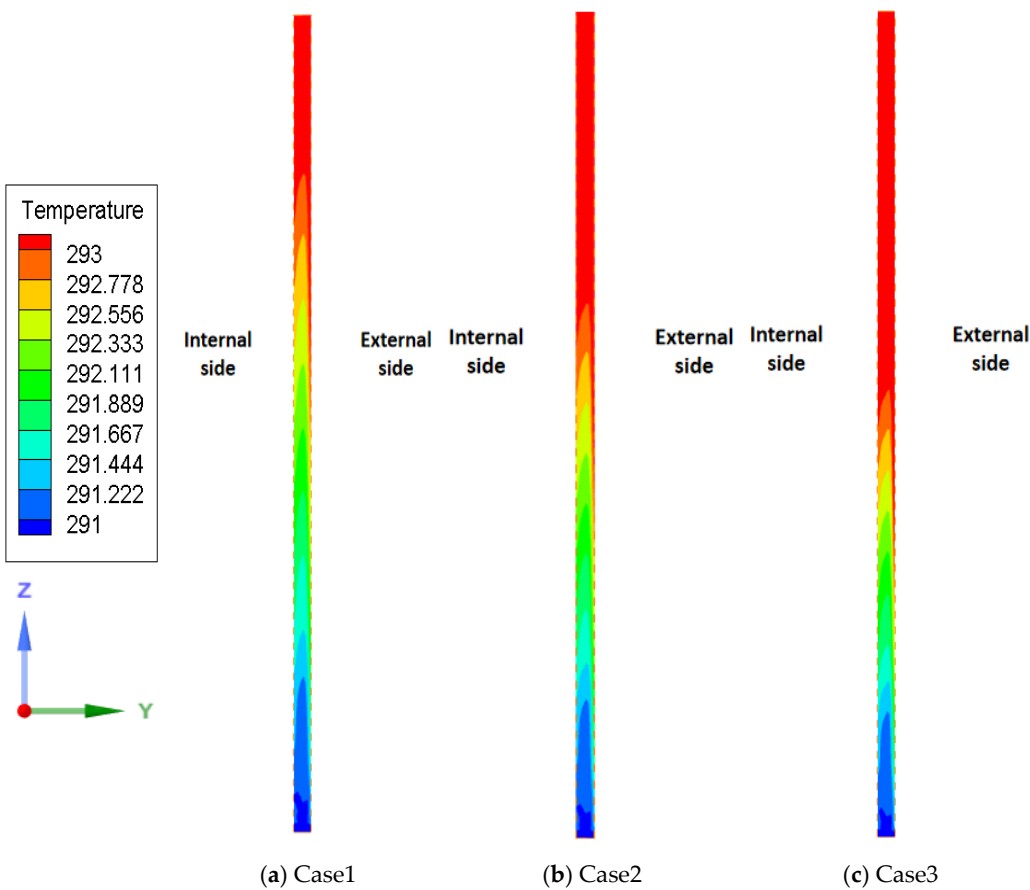

(**a**) Case1　　　　　　(**b**) Case2　　　　　　(**c**) Case3

**Figure 8.** Temperature distribution at X = 0.4 m.

### 4.2. The Impact of Supply Water Temperature

With solar intensity at 600 W/m$^2$ and water flow velocity of 0.033 m/s, the effect of supply water temperature (285 K, 291 K and 298 K) on the thermal characteristics is compared. Figure 9 shows the temperature distribution at Y = 0.01 m.

It can be seen from Figure 9 that the average temperature of the fluid in the window cavity is lower when water at a lower temperature is introduced. However, it can also be observed that with the increase in supply water temperature, the water temperature difference between the top and the bottom decreases; they are 7 K, 5 K and 4 K, respectively, for Cases 4, 2 and 5. Under the same solar condition, the convective heat exchange between the fluid and the glazing surface decreases with the increase in supply water temperature, as do the temperature rise and water heat gain.

The average temperature of each boundary surface and the water heat gain are listed in Table 9. As the supply water temperature increases, the temperature difference between the inlet and the outlet decreases; they are 3.8 K, 2.7 K and 1.7 K for the three cases. Correspondingly, the water heat gain is reduced; the minimum one is 162.71 W for Case 5 with a supply water temperature of 298 K. The surface temperatures of glazing panes are affected by heat exchange with the water stream, and thus a rise in surface temperature with the increase in supply water temperature is observed.

**Table 9.** Average temperatures at boundary surfaces and water heat gain.

| Case | $T_{inlet}$ (K) | $T_{outlet}$ (K) | $Q_w$ (W) | $T_{inner-glazing}$ (K) | $T_{outer-glazing}$ (K) |
|---|---|---|---|---|---|
| 4 | 285 | 288.60 | 287.20 | 286.96 | 287.69 |
| 2 | 291 | 293.71 | 229.84 | 292.41 | 293.10 |
| 5 | 298 | 299.72 | 162.71 | 298.76 | 299.39 |

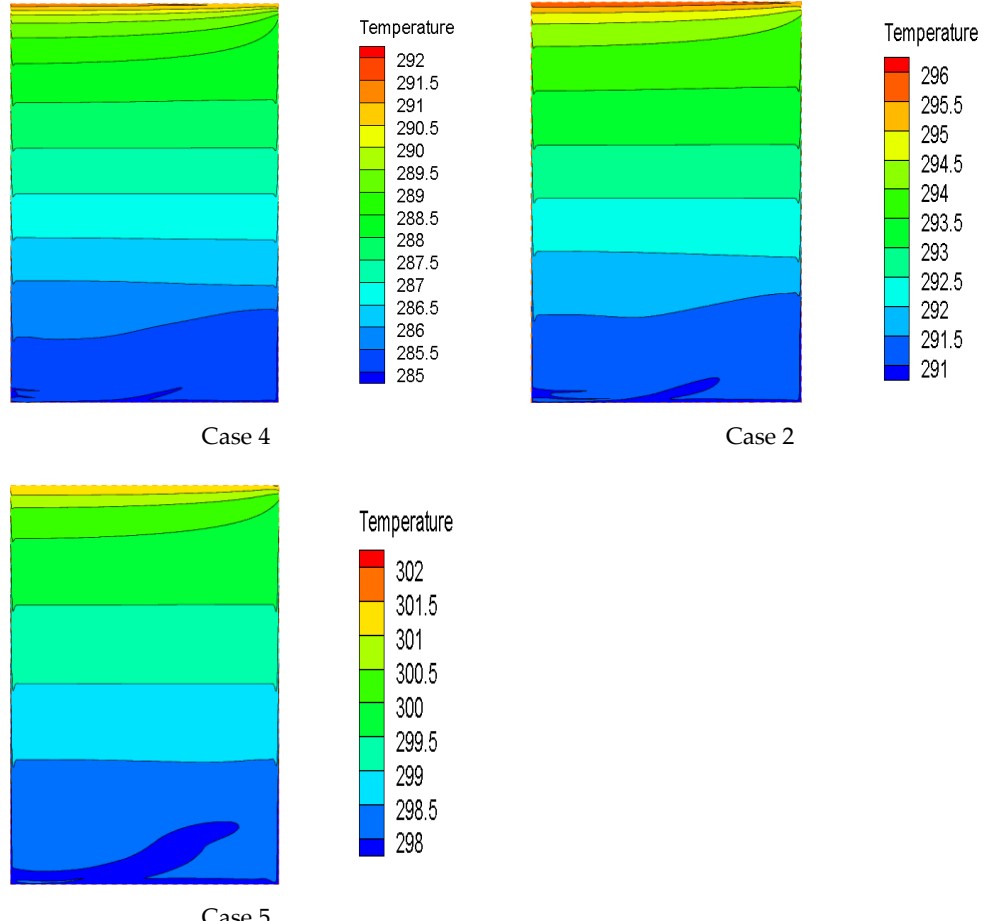

Figure 9. Temperature distribution at Y = 0.01 m.

In summer application, the lower the supply water temperature, the larger the water heat gain. Meanwhile, thermal energy can be effectively removed to reduce heat release to room space through the interior glazing pane. However, in practical application, the temperature of supply water in the municipal pipe network is affected by the outdoor environment. Therefore, it is necessary to make good insulation to improve the thermal efficiency and energy-saving performance of the water flow window.

Figure 10 shows the velocity distribution at the cross-section of Y = 0.01 m. From the perspective of global distribution, the velocity has the largest value near the inlet and the outlet, and the velocity distribution in the cavity shows a similar trend. Inlet vortices are observed at the bottom of the window cavity. Moreover, a larger velocity near the window inlet is observed with the increase in supply water temperature.

A summary of the average water flow velocities near the window surfaces (Y = 0.017 m, Y = 0.003 m) and in the middle (Y = 0.01 m) is given in Table 10. The average flow velocity in the cavity is hardly affected by the variation of supply water temperature; it is about $9.41 \times 10^{-4}$ m/s for all three cases. The average velocity at the different cross-sections varies in the same pattern with the minimum value at Y = 0.01 m and the maximum one at Y = 0.017 m.

Table 10. Average velocity of fluid in cavity along the vertical direction.

| Case | $V_{Volume}$ (m/s) | $V_{y=0.003m}$ (m/s) | $V_{y=0.01m}$ (m/s) | $V_{y=0.017m}$ (m/s) |
|---|---|---|---|---|
| 4 | 0.000941 | 0.000696 | 0.000037 | 0.002706 |
| 2 | 0.000941 | 0.000520 | 0.000044 | 0.002859 |
| 5 | 0.000941 | 0.000201 | 0.000050 | 0.003064 |

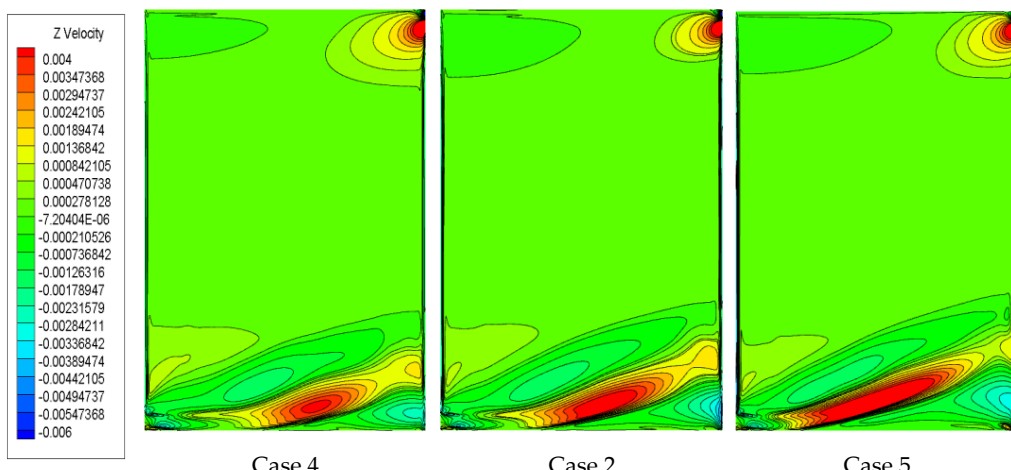

**Figure 10.** Velocity distribution at Y = 0.01 m.

At the cross-section of Y = 0.003 m, the water stream extracts thermal energy from room space through the interior glass pane, and its temperature is thus raised. Under the influence of buoyancy-driven force, the water stream flows upward, and the streamline distribution is dominantly uniform, as shown in Figure 11. This is especially true for Case 4 with lower supply water temperature and larger temperature rise. Local vortices at different scales are observed for Cases 2 and 5 near the bottom zone because of the reduced temperature rise. The uniform velocity distribution at Y = 0.017 m is largely related to the solar radiation, under which, the temperature rise of the water stream near outer glazing is higher by absorbing large amounts of solar thermal energy, resulting in the uniform upward flow. However, at Y = 0.01 m, the fluid temperature rise is of much smaller magnitude, and together with the influence of the inlet vortex, the streamline is partly disordered near the bottom.

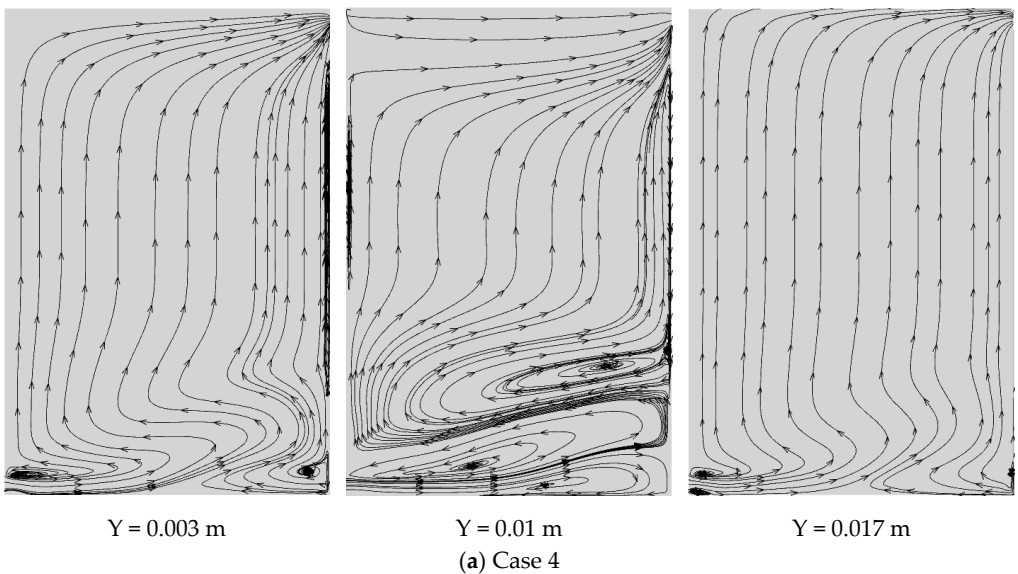

Y = 0.003 m             Y = 0.01 m             Y = 0.017 m

(**a**) Case 4

**Figure 11.** *Cont*.

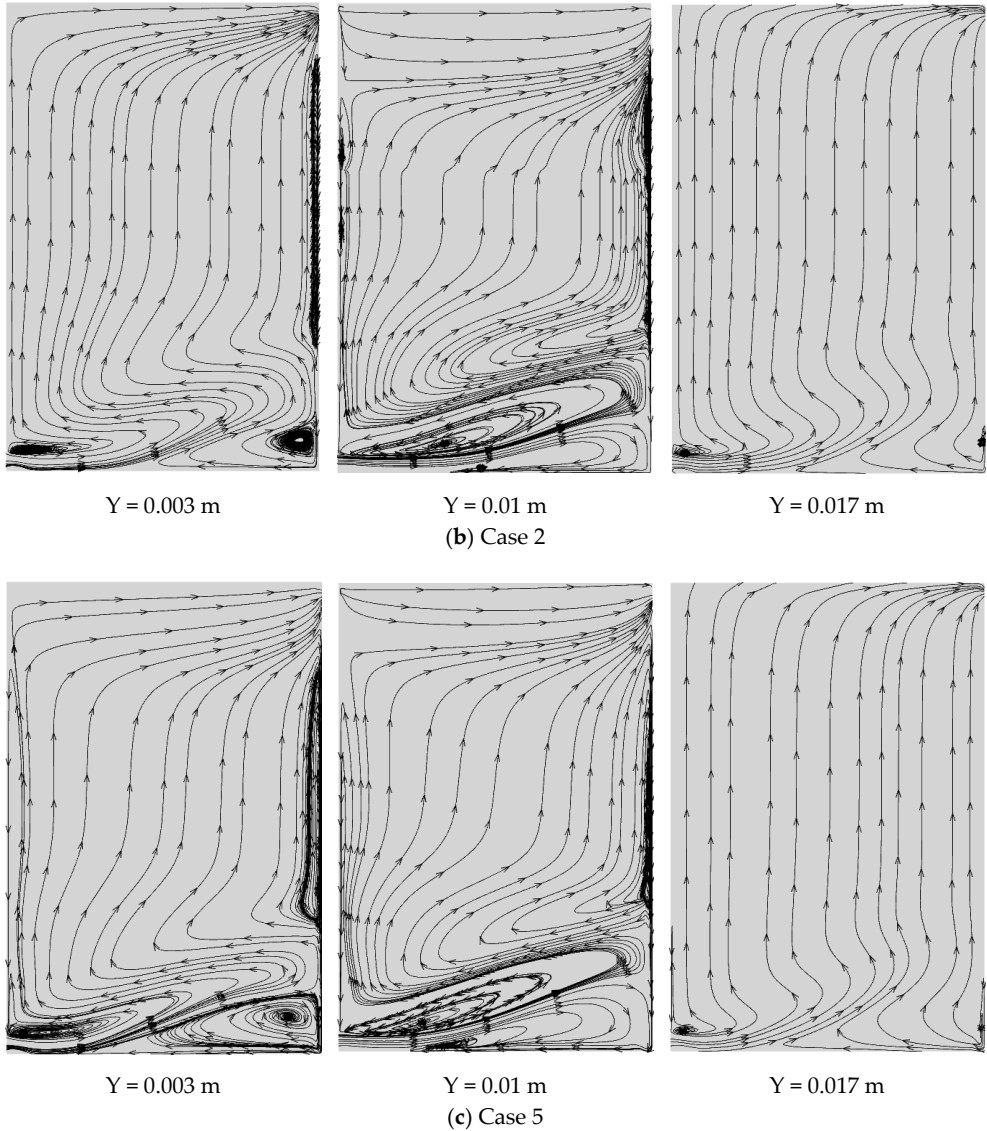

| Y = 0.003 m | Y = 0.01 m | Y = 0.017 m |

(**b**) Case 2

| Y = 0.003 m | Y = 0.01 m | Y = 0.017 m |

(**c**) Case 5

**Figure 11.** Streamline at the cross-sections of Y = 0.003 m, Y = 0.01 m and Y = 0.017 m.

As given in Table 10, with the increase in supply water temperature, the velocity at Y = 0.003 m decreases gradually, while it increases at the cross-section of Y = 0.01 m. From Figure 11, it can be observed that at Y = 0.003 m, the increase in supply water temperature increases the vortex range as mentioned above, and thus the streamline becomes chaotic. The reason is that the buoyancy-driven force is reduced at a smaller temperature rise. However, at the cross-section of Y = 0.01 m, due to the reduced temperature difference between the fluid near the window surface and at the intermediate, the backflow vortex formed thereby becomes flat, and the average velocity is then increased. At the cross-section of Y = 0.017 m, the fluid temperature is affected directly by solar radiation other than the temperature difference with the environment. Therefore, the difference in temperature rise is not prominent among the three cases. The streamline is then uniform, and the average velocity changes little as given in Table 10.

### 4.3. The Impact of Water Supply Velocity

Taking the solar intensity of 600 W/m$^2$ and the supply water temperature of 291 K, the influence of water supply velocity is analyzed. Figure 12 shows the temperature distribution at the cross-section of Y = 0.01 m with supply water velocities of 0.017 m/s, 0.033 m/s, 0.083 m/s and 0.17 m/s.

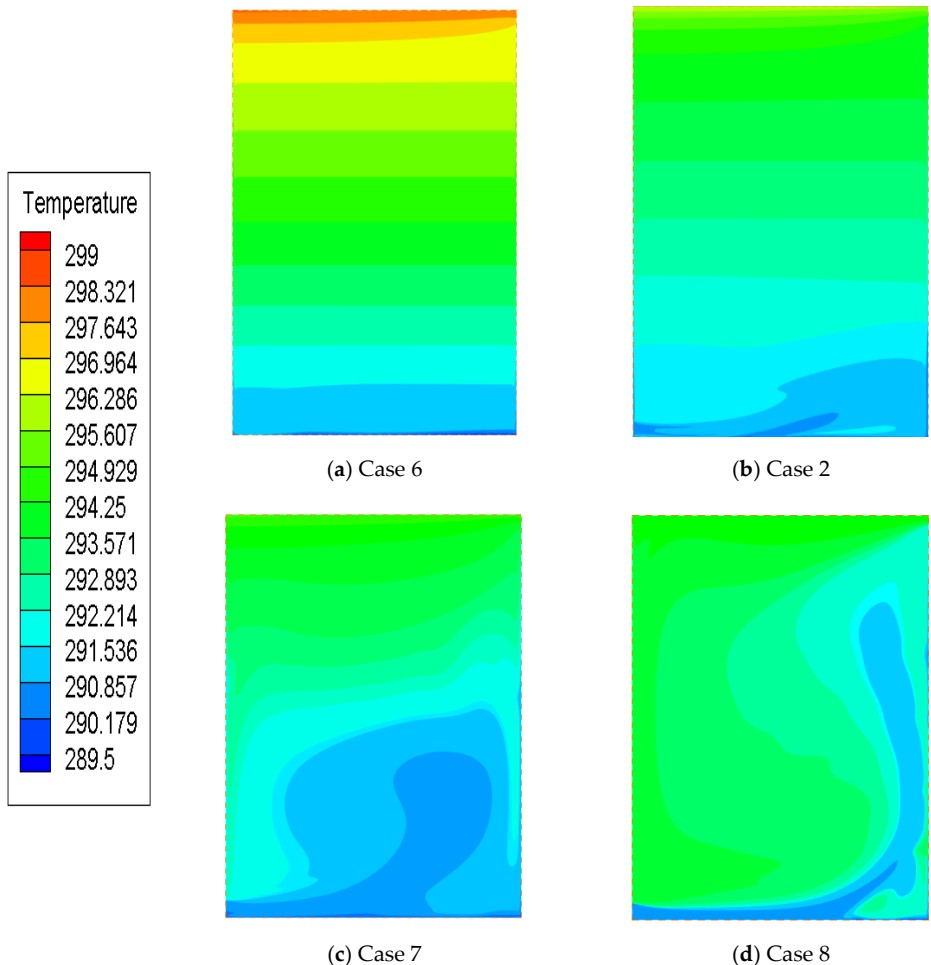

**Figure 12.** Temperature distribution at Y = 0.01 m.

At the smaller flow velocity, the water stream takes more time to pass through the window cavity; its temperature rise is thus larger as can be observed for Case 6. In Figure 12a,b, the water temperature is evenly layered and increases from the bottom to the top under the influence of buoyancy. However, for Cases 7 and 8 with larger velocity, the average temperatures are at a lower level, and nearly horizontal stratification is formed, as can be observed in Figure 12c,d. The reason is that the temperature distribution is largely affected by the incoming forced flow under the high-velocity condition. In Case 8, the high-velocity water stream reaches the side wall of the cavity, partly exits the window cavity through the outlet and partly diffuses along the opposite direction. Horizontal temperature stratification is thus formed.

Table 11 lists the average temperatures of the water layer and the glazing surfaces, as well as the water temperature at the window outlet and the corresponding water heat gain. In Case 6 with the smallest water supply velocity, the water temperature at the outlet is the highest, and the temperature rise between the inlet and the outlet reaches 6.4 K. As the velocity increases, the temperature rise decreases. Meanwhile, the water heat gain varies. With the increase in water flow velocity, the water heat gain increases first and then decreases. Comparing Cases 7 and 8, a small decrease from 241.22 W to 240.66 W is found. This indicates that a proper increase in water flow velocity can enhance the convective heat transfer and increase the heat gain of the water stream. However, further increase in water flow velocity may cause a reduction in water heat gain. This is also true under the other conditions. For example, at a solar intensity of 600 W/m$^2$ and supply water temperature of 298 K, the water heat gain rises as the water supply velocity increases from 0.017 m/s to 0.17 m/s, while a decrease is found by further increasing the velocity to 0.32 m/s.

**Table 11.** The average temperature of each boundary surface at different inlet velocities.

| Case | $T_{Volume}$ (K) | $T_{outlet}$ (K) | $Q_w$ (W) | $T_{inner-glazing}$ (K) | $T_{outer-glazing}$ (K) |
|---|---|---|---|---|---|
| 6 | 294.25 | 297.40 | 214.27 | 294.63 | 295.01 |
| 2 | 292.66 | 294.47 | 229.84 | 292.77 | 293.60 |
| 7 | 291.80 | 292.45 | 241.22 | 291.82 | 292.41 |
| 8 | 291.67 | 291.72 | 240.66 | 291.70 | 292.24 |

The velocity field of the four cases at the cross-section of Y = 0.01 m is shown in Figure 13. In Cases 2 and 6, the supply water stream at a smaller velocity absorbs a larger amount of thermal energy and is heated to a higher temperature. Under such condition, the velocity field is dominated by buoyancy-driven flow, and the streamline is evenly distributed, as shown in Figure 13a,b.

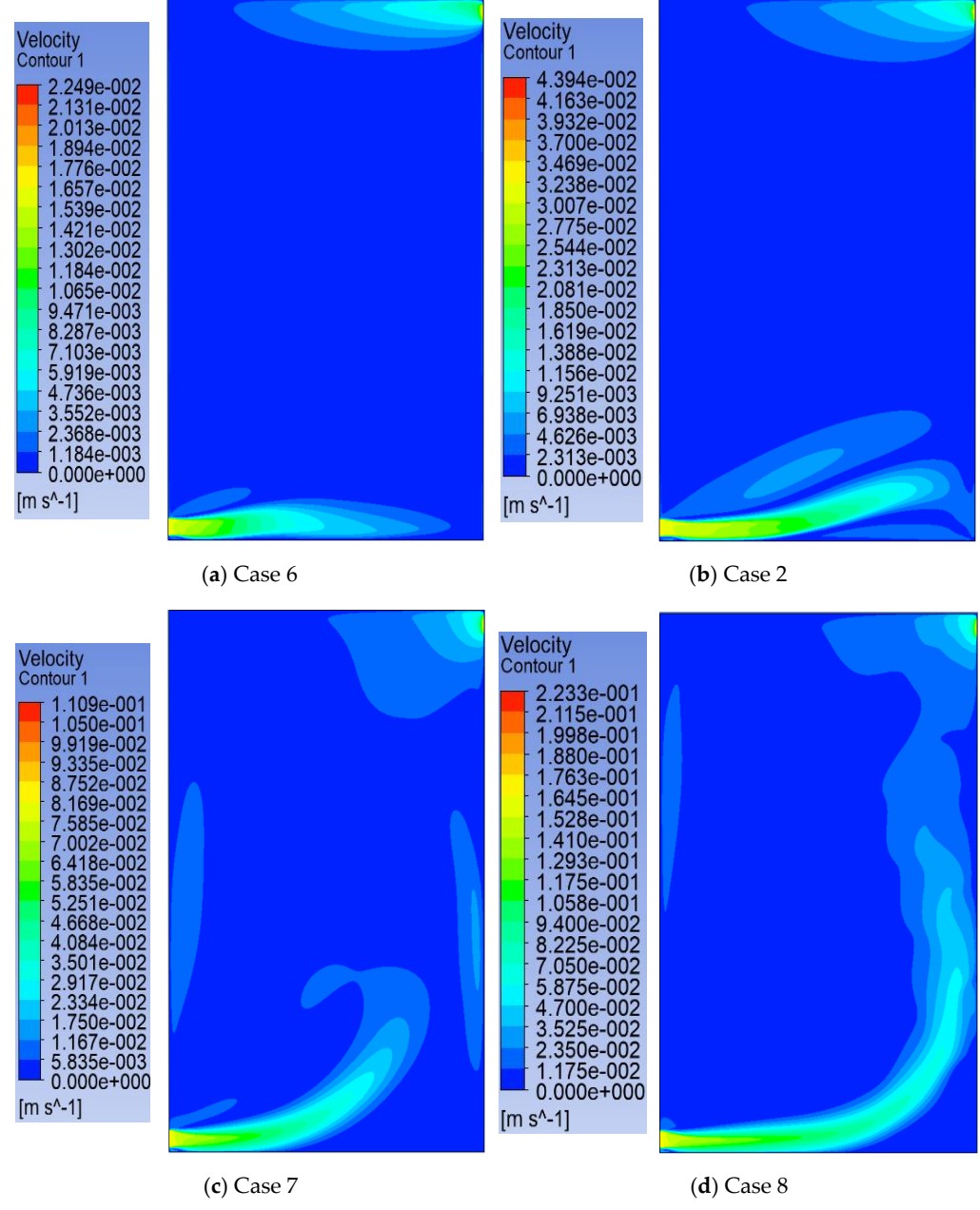

(**a**) Case 6      (**b**) Case 2

(**c**) Case 7      (**d**) Case 8

**Figure 13.** Comprehensive velocity distribution at Y = 0.01 m.

For Case 7, the fluid flows upward, and high-speed areas appear locally near the side surfaces. From Figure 14c, it can be observed that this is related to the water flow diffusion toward the side surfaces and the resistance of the solid wall. Similarly, because of the water diffusion and the resistance from the side surfaces as shown in Figure 14d, a long strip of a high-velocity zone appears on the left side surface in Case 8, as illustrated in Figure 13d.

In all four cases, an inlet vortex is observed at the bottom of the window cavity. However, differences in streamline distribution are also observed. As shown in Figure 14a,b, the streamlines at the cross-sections of Y = 0.003 m and Y = 0.017 m show a uniformly upward flow. This is because the water stream near the glazing surfaces is heated to a higher temperature after absorbing thermal energy from the indoor and outdoor environments. It then flows upward under the buoyancy-driven force. However, in Case 6, the streamline shows a downward flow at Y = 0.01 m, and additional vortices appear at the bottom. This is caused by the larger temperature difference between the fluid near the glazing surface and in the middle. For Cases 7 and 8, affected by the high-velocity incoming flow, forced flow is dominant, and the streamline is partly disordered. As the water flow velocity increases, the inlet vortex increases accordingly. The formation of the vortex will unfavorably affect heat transfer and thermal removal.

In real application, consideration should be given to the influence of the water supply velocity on the water heat gain and indoor thermal environment. As the water flow velocity increases, larger amounts of thermal energy can be extracted to maintain the stability of the indoor thermal environment and reduce the indoor cooling load. At the same time, the high-temperature water at the window outlet can be delivered to the building's hot water system, thereby achieving the goal of building energy saving. However, the velocity should be determined properly; providing an excessively large flow velocity is not only ineffective in thermal performance improvement but also requires larger amounts of pump energy consumption.

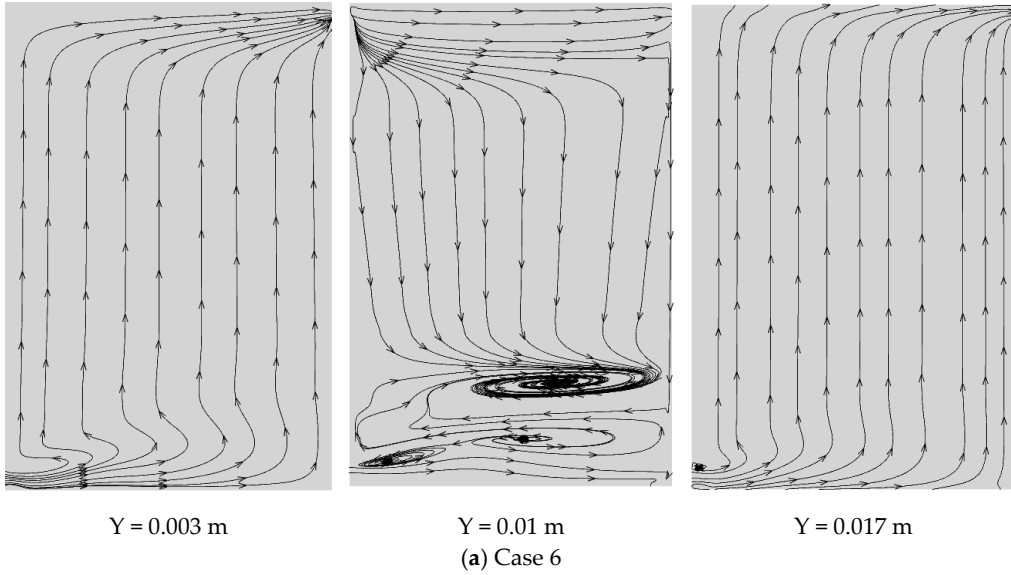

| Y = 0.003 m | Y = 0.01 m | Y = 0.017 m |

(**a**) Case 6

**Figure 14.** *Cont.*

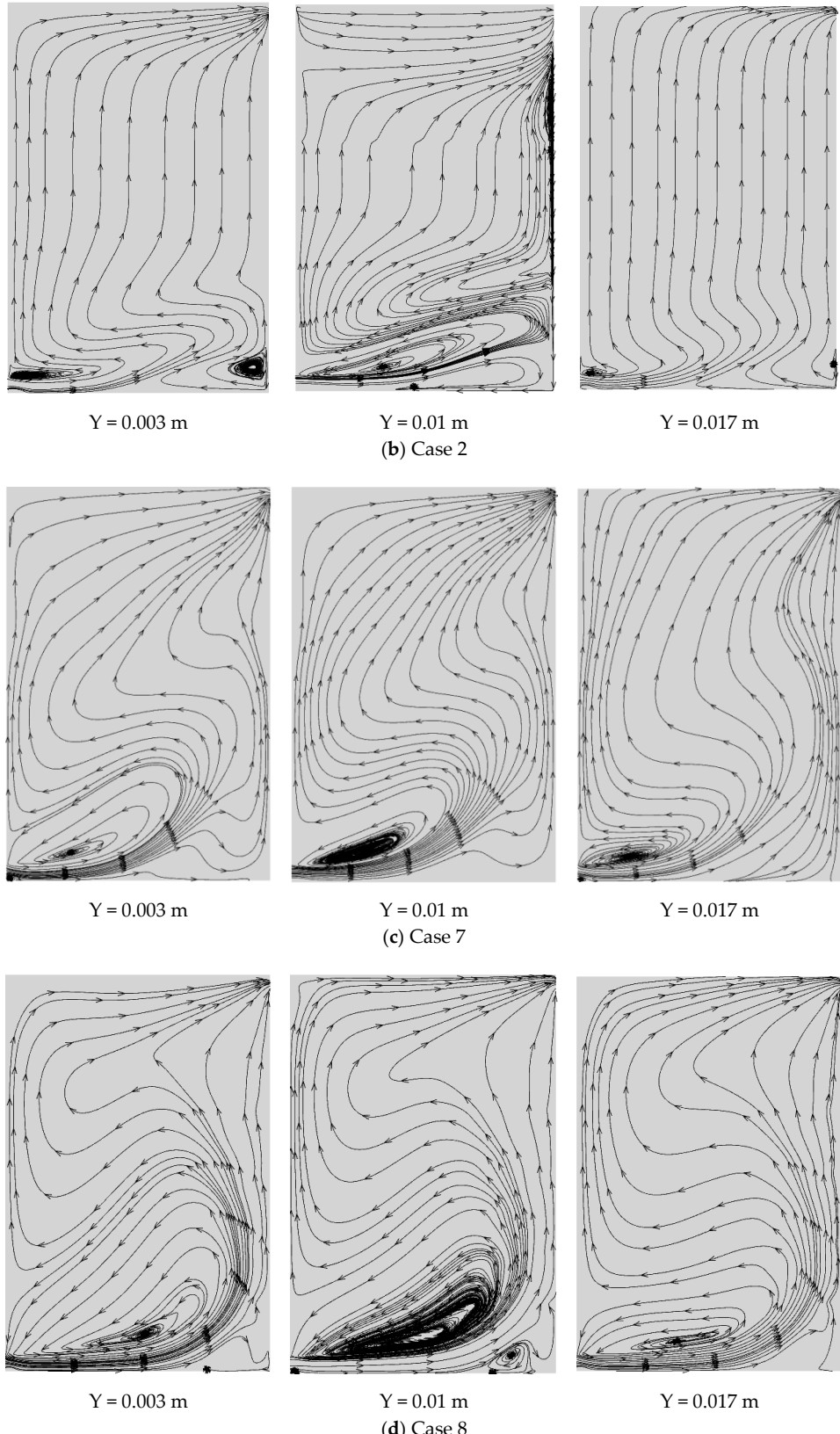

| Y = 0.003 m | Y = 0.01 m | Y = 0.017 m |
| :---: | :---: | :---: |
| | (**b**) Case 2 | |

| Y = 0.003 m | Y = 0.01 m | Y = 0.017 m |
| :---: | :---: | :---: |
| | (**c**) Case 7 | |

| Y = 0.003 m | Y = 0.01 m | Y = 0.017 m |
| :---: | :---: | :---: |
| | (**d**) Case 8 | |

**Figure 14.** Streamline at the cross-sections of Y = 0.003 m, Y = 0.01 m and Y = 0.017 m.

## 5. Conclusions

This paper analyzed the thermal characteristics of a novel green facade water flow window with CFD simulation. The correctness of the simulation method was successfully

validated. The impacts of solar intensity, water supply temperature and velocity on the temperature and velocity distribution, as well as thermal performance, were explored. The major findings are summarized as follows:

(i)   With the increase in solar intensity, the temperature rises, and the heat gain of the water stream increases. At the same time, local vortices are formed in the cavity because of the temperature difference in the water stream. With larger solar intensity, the number of vortices is increased, and the location is closer to the bottom. The existence of a vortex unfavorably influences the increase in heat transfer efficiency and should be eliminated in real application.

(ii)  At higher inlet water temperature, the thermal extraction capacity of the water stream is found weakened. At the same time, the varied water temperature rise also affects the state of the local vortex. With reduced supply water temperature, the larger temperature rise enhances the buoyancy-driven upward flow, and the scale of the vortex is thus reduced. Consequently, thermal collection performance can be improved.

(iii) In the low-velocity range, the increase in flow velocity reduces the temperature rise but increases the useful water heat gain. Further increase in the supply water velocity may cause decreases in water heat gain. Therefore, it should be determined carefully to obtain the maximum water heat gain with low pump energy consumption.

This study provides a better understanding of the temperature and velocity distribution characteristics as well as the heat transfer mechanism in the flow field of the water flow window, and suggestions for practical application are put forward to improve its energy-saving performance. With an improved structural and operational design, the enhanced thermal collection will contribute to energy saving in both air-conditioning and water-heating systems and is beneficial for carbon emission reduction from the processes of power generation to equipment operation.

**Author Contributions:** Y.L.: investigation, supervision, methodology. S.C.: writing—original draft, formal analysis, data curation, software. C.L. (Can Liu): writing—review and editing, formal analysis, data curation. J.L.: writing—review and editing, validation. C.L. (Chunying Li): data curation. H.S.: formal analysis, investigation. All authors have read and agreed to the published version of the manuscript.

**Funding:** This research was funded by National Natural Science Foundation of China (NSFC) (51808449); the Key Research Project, Science and Technology Bureau of Chengdu (2019-YF05-02174-SN); and the Key Project of Xihua University (Z201048).

**Data Availability Statement:** Not applicable.

**Conflicts of Interest:** The authors declare no conflict of interest.

## Nomenclature

| Nomenclature | | Greek Letters | |
|---|---|---|---|
| $C$ | specific heat capacity, J/(kg·K) | $\rho$ | density, kg/m$^3$ |
| $\vec{g}$ | gravitational acceleration, m/s$^2$ | $\mu$ | molecular viscosity, kg/(m·s) |
| $H$ | heat transfer coefficient, W/(m$^2$·K) | $V$ | velocity, m/s |
| $I$ | unit tensor | *UFR* | under-relaxation factor |
| $k$ | thermal conductivity, W/(m·K) | *SIMPLEC* | SIMPLEC-Consistent |
| $\dot{m}$ | mass flow, kg/s | *DO* | discrete ordinates model |
| $N$ | number of data points | **Subscripts** | |
| $P$ | working pressure, Pa | *exp* | experimental |
| $Q$ | heat, W | *indoor* | indoor environment |
| $S$ | heat source, W/m$^3$ | *inlet* | window inlet |
| $t$ | time, s | *outdoor* | outdoor environment |
| $\overset{=}{T}$ | stress factor | *outlet* | window outlet |
| $T$ | temperature, K | *sim* | simulation |
| $\vec{u}$ | flow velocity vector, m/s | *w* | water |

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
