# Peer review of "Thermal Characteristics Simulation of an Energy-Conserving Facade: Water Flow Window"

_sustainability, doi:10.3390/su14052737_

Round 1

Reviewer 1 Report

Dear Authors,

I've found the paper interesting and the results appropriate. Anyway, I suggest publishing it after minor changes.

Please check the dimensions of Equation n.5. They look inconsistent.

Generally, be consistent with Nomenclature by using the same dimensions for all over the text. I.E. use always °C or K for the temperature.

Authors don't mention how the water exiting the windows is managed. Does the water exiting the window pass through a heat exchanger and then recirculated or it is wasted out?
In this case how the authors deal with the water waste?

It is not clear where the window surface temperature are evaluted. Please provide a scheme where this points are clearly visibile.

Fig.5 For a better understanding, please specify the reference system adopted and the position of the plane where the contour maps are evaluated. 

Author Response

Thank you very much for the comment. For detailed revisions, please see the attachment.

Reviewer 2 Report

I have read with interest the article. "Thermal Characteristics Simulation of An Energy Conserving 2 Facade: Water Flow Window". 

The authors use ANSYS FLUENT software for CFD simulation was completed to explore the thermal characteristics in a water flow window at a micro-scale. 

This is a well-documented and well-written manuscript that can be published in the journal of Sustainability, still, the figures and the tables I think should be better arranged. The paper is very comprehensible and very abundant in data, also the conclusion reflects the research work.

Author Response

(The authors gave the same response as above.)

Reviewer 3 Report

This paper presents a series of CFD calculations for the so called water flow window. Moreover, the set of governing equations is presented. Experimental data are also given and used in order to validate commercial CFD code. Finally, some quantitative conclusions are drawn. Some of them are trivial like: "With the increase of solar intensity, the temperature rises and the heat gain of the water stream increases." or "At higher inlet water temperature, the thermal extraction capacity of the water stream is found weakened". Nevertheless, the problem is interesting and important form the energy saving point of view. What concerns me most, is the "microscale" term used throughout the paper. In my opinion all calculations are in the macroscale. Otherwise macroscopic conservation equation cannot be utilised. It is no clear to me what is the original contribution of the authors. This should be clearly stated in the introduction. Next, I wonder if the authors treated water as a compressible medium, as indicated by equations (1)-(5). Or perhaps was the simulation carried out using the Boussinesq approximation? I think that the paper needs serious revisions. The following problems should be also properly addressed: 1. line 212 "laminar or turbulent flow models were used". There are no information on wall function, turbulence intensity whatsoever 2. page 2: wrong unit of 'g'. What is more, it is not the "gravity velocity vector" 3. page 2: not SIMPLIC but SIMPLEC 4. page 4: eq. (1) is incorrect. In the first (transient) term it should be Greek 'rho' rather than 'p' 5. page 4: eq. (2) is incorrect. It should be divergence of the stress tensor rather than gradient 6. page 4, line 96-o7. 'T' is not the 'stress factor' but 'stress tensor' 7. page 4, eq. (3). If you start denote tensors with double overline the you should do the same for the unit tensor 8. page 4, eq. (4). The first term on the right hand side: it should be divergence rather than gradient 9. page 4, line 104: temperature is expressed in Kelvins not in deg. Celsius 10. page 4, eq. (5): the units are inconsistent. Heat transfer coefficient is not expressed in m/s as the right hand side may suggests 11. page 6, table 2: the unit (kg/s) of 'inlet velocity' is incorrect or possibly it is rather mass flow rate instead 12. line 151, 202: "mesh structures were encrypted" encrypted?

Author Response

(The authors gave the same response as above.)

Round 2

Reviewer 3 Report

The paper has been mostly corrected. However, the authors themselves admitted that the calculations were treated as incompressible. Thus, equation (1) should be in an incompressible form. The same applies to the stress tensor (3). An additional term related to the Boussineq approximation should be added to equation (2). Otherwise the system of equations (1)-(4) is not a closed and cannot be solved in this form. As for the incorrect units in equation (5), one of the reviewers also noted this problem. Whether the equation is empirical or not should not affect its physicality and therefore the correctness of the units.

Author Response

Thank you very much for the comment.Please see the attachment.
